# Overcoming Graft Rejection in Induced Pluripotent Stem Cell-Derived Inhibitory Interneurons for Drug-Resistant Epilepsy

**DOI:** 10.3390/brainsci14101027

**Published:** 2024-10-16

**Authors:** Cameron P. Beaudreault, Richard Wang, Carrie Rebecca Muh, Ashley Rosenberg, Abigail Funari, Patty E. McGoldrick, Steven M. Wolf, Ariel Sacknovitz, Sangmi Chung

**Affiliations:** 1School of Medicine, New York Medical College, Valhalla, NY 10595, USA; beaudrec@upstate.edu (C.P.B.); rwang5@student.nymc.edu (R.W.); arosenbe14@student.nymc.edu (A.R.); asacknov2@student.nymc.edu (A.S.); 2Department of Neurosurgery, Westchester Medical Center, Valhalla, NY 10595, USA; 3Department of Neurosurgery, SUNY Upstate Medical Center, Syracuse, NY 13210, USA; 4Division of Pediatric Neurology, Maria Fareri Children’s Hospital, Valhalla, NY 10595, USA; 5Division of Pediatric Neurology, Boston Children’s Health Physicians, Hawthorne, NY 10532, USA; 6Department of Cell Biology and Anatomy, New York Medical College, Valhalla, NY 10595, USA; 7Department of Neurosurgery, Brain Health Institute, Robert Wood Johnson Medical School, Rutgers University, Piscataway, NJ 08854, USA

**Keywords:** induced pluripotent stem cells, inhibitory interneurons, drug-resistant epilepsy (DRE)

## Abstract

Background: Cell-based therapies for drug-resistant epilepsy using induced pluripotent stem cell-derived inhibitory interneurons are now in early-phase clinical trials, building on findings from trials in Parkinson’s disease (PD) and Huntington’s disease (HD). Graft rejection and the need for immunosuppressive therapy post-transplantation pose potential barriers to more epilepsy patients becoming potential candidates for inhibitory interneurons transplantation surgery. Objectives: The present literature review weighs the evidence for and against human leukocyte antigen (HLA)-mediated graft rejection in PD and HD and examines the potential advantages and drawbacks to five broad approaches to cell-based therapies, including autologous cell culture and transplantation, in vivo reprogramming of glial cells using viral vectors, allogeneic transplantation using off-the-shelf cell lines, transplantation using inhibitory interneurons cultured from HLA-matched cell lines, and the use of hypoimmunogenic-induced pluripotent stem cell-derived inhibitory interneurons. The impact of surgical technique and associated needle trauma on graft rejection is also discussed. Methods: Non-systematic literature review. Results: While cell-based therapies have enjoyed early successes in treating a host of central nervous system disorders, the immunologic reaction against surgical procedures and implanted materials has remained a major obstacle. Conclusions: Adapting cell-based therapies using iPSC-derived inhibitory interneurons for epilepsy surgery will similarly require surmounting the challenge of immunogenicity.

## 1. Introduction

Drug-resistant epilepsy (DRE) is any form of epilepsy in which patients do not achieve adequate seizure control after two adequate trials of appropriately selected anti-seizure medications [1,2]. Uncontrolled seizures have a debilitating effect on morbidity and mortality, with the potential to cause neuropsychiatric effects ranging from depression to anxiety and increasing the risk of sudden unexplained death in epilepsy [2]. While epilepsy surgery can be curative, surgical candidacy depends on either the presence of focal lesions for resection or ablation or on identifying seizure foci using a combination of non-invasive and invasive electroencephalography recordings [3,4]. Important exceptions to candidacy include multifocal-onset seizures and seizure foci associated with the eloquent cortex or other highly functional zones, wherein resection would prove debilitating to quality of life [5,6]. Neuromodulation, including vagus nerve stimulation, deep brain stimulation, and responsive neurostimulation, help reduce seizure burden and, in some cases, offer complete seizure freedom; however, these procedures involve implants which require follow-up care with a neurologist for programming and carry a risk of revision surgeries for infection, device failure, or due to inadequate seizure suppression, as well as regular follow-up surgeries for battery replacement. Due in part to the heterogeneity of patients’ etiologies for their epilepsy, additional novel therapies are needed to control seizures and improve quality of life for epilepsy patients that are not candidates for current therapies or who continue to have uncontrolled seizures while receiving therapy [7]. 

An emerging cell-based therapy for DRE now undergoing clinical trials is the transplantation of postmitotic cortical inhibitory interneurons derived from induced pluripotent stem cells (iPSCs) into seizure foci or the neocortex [8]. These gamma-aminobutyric acid (GABA) secreting inhibitory interneurons are the primary source of synaptic inhibition in the hippocampus and neocortex, and their loss is consistently reported in seizure foci [9,10,11]. Transplantation of inhibitory interneurons derived from human pluripotent stem cells (hPSCs) reduced spontaneous recurrent seizures in rodent models of temporal lobe epilepsy [12,13,14,15] and attenuated the development of epilepsy after provoked episodes of status epilepticus [13]. With clinical trials underway, the challenge of maintaining long-term graft viability and avoiding chronic rejection remains. The aim of the present literature review is to evaluate various strategies for modulating the host immune response to inhibitory interneuron transplantation. 

### 1.1. Central Nervous System Immune Privilege

While immunological access to the central nervous system (CNS) is tightly regulated, the brain is not immunologically privileged [16]. During homeostasis, T cells patrol CNS-associated border regions and can breach the glia limitans if activated by antigen-presenting cells [17]. Dural lymphatic vessels facilitate the drainage of CNS antigens to peripheral lymph nodes [18], aided by the glymphatic system of cerebrospinal fluid (CSF) drainage [19]. The glymphatic system also provides a gateway through which activated T cells readily access the CNS. Together, these findings describe a clear mechanism for the cell-mediated immune rejection of engrafted cells.

Evidence from clinical trials of cell therapies for Huntington’s disease (HD) and Parkinson’s disease (PD) supports the concern for graft rejection. In a clinical trial investigating the transplantation of fetal tissue in PD patients, the weaning and discontinuation of cyclosporine 6 months post-transplant were associated with a loss of graft efficacy in ameliorating disease symptom burden [20]. Although cases of cell graft survival have been reported more than ten years after transplantation, graft rejection is thought to play a role in the low survival rate of engrafted midbrain dopaminergic neurons seen in numerous cell therapy trials for PD [21]. As such, the TRANSEURO trial protocol, the largest ongoing trial of fetal tissue transplants in PD patients, maintains immunosuppressive therapy for one year post-transplant [22]. Similarly, in HD patients, a pilot study investigating the transplantation of medial ganglionic eminence (MGE)-derived cells demonstrated degrees of alloimmunization and graft rejection [23,24], and a phase II multicenter trial based on the previous pilot study reported many serious adverse events for which graft rejection was implicated [24]. Graft rejection, primarily driven by the absence of histocompatibility in human leukocyte antigen (HLA) between graft and host [25], remains a challenge for cell-based therapies for CNS disorders, including DRE.

### 1.2. Autologous Derivation and Culture

The most direct and sureproof way of avoiding immune rejection for implanted inhibitory interneurons from HLA-mismatch is to culture the interneurons from induced pluripotent stem cells that were derived from the patient. This approach, considered by some to be financially and logistically prohibitive with a per-patient estimate of USD 300,000–800,000 to generate a clinical-grade iPSC cell product [26,27,28], has also generated innovative potential solutions, including single-tube reprogramming and culture. While the logistics of such an approach are still under debate, autologous iPSCs have been successfully cultured to substantia nigra-like neurons and implanted with symptom stabilization in a case report of PD [29]. Limitations of the study included its single case of use and lack of blinding as the patient and the rater were not blinded to the intervention. The time frame of clinical changes and positron emission tomography (PET) signal improvements was otherwise consistent with a time frame of the reinnervation of putamen neurons without requiring the usage of immunosuppressants and have led to four clinical trials assessing the viability of iPSC use in PD [30,31,32,33].

While iPSCs are currently being utilized in a variety of disease states of different organ systems, autologous iPSC clinical trials in most systems are less frequent due to the aforementioned challenges in cost and labor despite documented success in their use [26,27,28,34], and iPSC usage in epilepsy is similarly limited to few clinical trials, one of which is examining the usage of iPSC-derived exosome nasal droplets in refractory epilepsy [35]. Similarly to the explosion of clinical trials of usage of iPSCs in PD, increasing case reports and series examining the outcomes of iPSC usage in epilepsy may increase the willingness to initiate trials on a larger scale. A path in reducing autologous iPSC derivation and cultivation may also be learned from the success seen in reducing the costs of chimeric antigen receptor T (CAR-T) cell therapy, wherein an Indian company was able to produce a CAR-T cell therapy at one-tenth the price of comparable American companies [36].

### 1.3. Viral Vector Reprogramming

Reprogramming a patient’s own cells to become inhibitory interneurons in vivo offers a promising approach for treating DRE. One technique involves using viral vectors to induce the expression of neurogenic transcription factors, such as achaete–scute family bHLH transcription factor 1 (ASCL1) and distal-less homeobox 2 (DLX2), enabling the direct conversion of glial cells into functional GABAergic interneurons within the patient’s brain, thus bypassing the need for surgical interventions and cell transplantation [37]. One of the benefits of this approach, as aforementioned, is the autologous derivation and culture, minimizing the risk of immune rejection to enhance the long-term viability and integration of reprogrammed cells [37,38]. Additionally, this approach can be personalized to fit the specific genetic makeup of the patient, potentially leading to more effective and targeted treatments.

The reprogrammed cells, once integrated, establish GABAergic synapses with existing neurons, thereby contributing to the regulation of excitatory activity and reducing the frequency and severity of seizures. This has been demonstrated in mouse models of mesial temporal lobe epilepsy (MTLE), where reprogrammed interneurons significantly reduced chronic seizure activity and established functional inhibitory networks within the hippocampus [37,38,39,40]. In one study, the use of Moloney murine leukemia virus-based retroviruses to knock in ASCL1 and DLX2 in reactive glial cells in the hippocampus of kainic acid-induced MTLE mice resulted in the efficient reprogramming of these glial cells into GABAergic interneurons. These newly formed interneurons integrated into the existing neural circuits and significantly reduced both the number and duration of spontaneous recurrent seizures [37]. However, there are several challenges and drawbacks to this approach. The process of in vivo reprogramming using viral vectors is technically complex and requires the precise delivery and control of gene expression. Importantly, validating cortical inhibitory interneurons which have been generated from reprogrammed virus-infected cells is also unlikely to be achievable in vivo without an invasive procedure. There are also critical safety concerns related to the use of viral vectors, such as the potential for insertional mutagenesis, which could disrupt host genes and lead to unintended consequences like oncogenesis [40]. Additionally, the long-term outcomes of this therapy are still uncertain, with a need for further studies to understand the durability and stability of the reprogrammed cells and their functional integration into neural circuits. Regulatory and ethical challenges also pose significant hurdles, as gene therapy approaches must navigate a complex landscape of compliance and ethical considerations [37].

Viral vector reprogramming of autologous cells has shown success in other pathologies of the CNS. In the context of PD, researchers have successfully converted astrocytes into functional dopaminergic neurons by depleting the RNA binding polypyrimidine-tract-binding protein (PTB) with a lentivirus vector delivering an RNA-targeting PTB, demonstrating the potential of this approach to restore motor functions in disease models [40]. This technique also highlights the importance of astrocyte plasticity and regional specificity in neuronal conversion, which could be leveraged to treat various neurodegenerative disorders.

### 1.4. Allogeneic Transplantation in Neurodegenerative Disease

Graft survival up to 24 years after the transplantation of embryonic dopaminergic neurons has been reported in PD patients treated in open-label studies [41,42], with most engrafted neurons remaining healthy throughout the remainder of a patient’s lifespan. Three double-blinded, sham-controlled trials of cell-based therapy for PD were completed in the early 2000s [20,43,44], of which two trials used hESCs to overcome rejection due to their relatively immune privileged identity. One trial used cyclosporine A in the first 6 months’ postoperatively, and initial significant improvements in Unified Parkinson’s Disease Rating Scale (UPDRS) scores among transplant recipients between 6 and 9 months postoperatively were lost after 9 months’ follow-up [20]. Two trials used no immunosuppressive regimen [43,44], and while one study found symptom improvements to be limited to patients under the age of 60 years old, the other found no significant differences in mean motor scores or daily dose of levodopa or equivalents between pre-surgery baseline and 12 months’ postoperative follow-up among participants in the study and sham surgery groups. In addition, any improvements in UPDRS scores in the trials were found to be attributable to the effects of surgery itself, as the improvement was seen to the same magnitude and for the same 0–12-month postoperative period in the study and sham surgery participant groups. The results of these trials in light of their differing approaches to (or lack of) immunosuppression therapy suggest that chronic graft rejection may have been a factor. Indeed, autopsy analysis and immunostaining of the putamen in two patients of a trial [44] revealed the presence of cluster of differentiation (CD) 3+ and HLA class II antigen-presenting cells along graft tracts and peri-vascular areas, providing further evidence for alloimmunization. 

The multicentric intracerebral grafting in a Huntington’s disease (MIG-HD) trial in Europe examined the efficacy of fetal ganglionic eminence engraftment into the bilateral striatum for Huntington’s disease patients, wherein a documented case of graft rejection was reported in a multiparous woman participating in the study [24], presenting fourteen months post-transplant with rapid weight loss with worsening choreic symptoms and gait disturbances, causing falls. Magnetic resonance imaging and PET studies suggested acute encephalitis, and blood and CSF samples revealed immunoglobulin G (IgG) directed against HLA-A*3 and HLA-A*11, matching the haplotype of the two fetal tissues engrafted into the patient’s right and left striatum, respectively. No such anti-HLA-A IgG antibodies were detected on analysis of the patient’s serum collected and frozen before surgery, indicating de novo alloimmunization. Four other patients in the trial also showed alloimmunization on serum analysis, but without correlating imaging or clinical signs or symptoms. Importantly, immunosuppressive therapy was utilized during this trial, consisting of cyclosporine A for one year as well as a multi-month taper of azathioprine and prednisolone. The graft rejection case presented within four months of discontinuing all immunosuppressive agents, and the patient’s condition improved once her anti-rejection regimen was resumed. Overall, the MIG-HD trial showed no clinical benefit to patients, and anti-HLA antibodies were found in 40% of patients [24], despite the year-long immunosuppressive regimen. These results, together with the results of clinical trials of cell therapy in PD, suggest that while immunosuppressive therapies may improve allograft acceptance in the CNS, the process of alloimmunization still occurs, potentially requiring patients to remain on anti-rejection medications chronically. Such a solution for DRE patients, if necessary, would be suboptimal at best. 

Due to ethical constraints in the use of hESC tissue for research and clinical applications [45], iPSC-derived grafts have been increasingly investigated for cell-based therapies, including postmitotic inhibitory interneurons grafting for epilepsy. Terminal differentiation of cell lines using cyclin-dependent kinase inhibitors such as PD 0332991 (Palbociclib) prevents tumorigenesis in iPSC-derived cell lines [46,47]. However, because these are allogeneic postmitotic cells, conventional immunosuppression (cyclosporine A or other such agents) in the immediate postoperative period is merited. 

### 1.5. Allogeneic Transplantation in Epilepsy

Utilization of allogeneic transplantation of off-the-shelf cell lines is now in phase I/II clinical trials [48] and represents a promising approach to cell-based therapy for epilepsy patients. Due to the risk of graft rejection, the trial protocol uses perioperative and postoperative immune-suppressing therapy, with the option to wean and discontinue therapy after one year follow-up [49]. Opportunistic infections, endocrinologic dysregulation, and metabolic syndrome, as well as increased risk of developing certain malignancies, are all risks undertaken by any patient on immunosuppressive therapy, and the main adverse events reported by Neurona investigators which are attributable to the trial itself have been due to patients’ immune-suppressing regimens [50]. Notably, although the two NRTX-1001 trial patients with the longest follow-up reportedly maintained graft-associated reductions in seizure frequency >95% at >1 year post-transplant, imaging-based follow-up is planned for up to two years post-transplant, and annual office visits supplemented by quarterly phone calls are planned for up to thirteen years post-transplant. These long-term follow-up studies will provide important evidence for or against the presence of chronic graft rejection in allogeneic cell lines of inhibitory interneurons.

Limitations of current allogeneic transplantation in epilepsy include the experimental nature of the procedure, with fewer evidence or ongoing trials [49] than in PD or HD [20,21,22,23,24,40,41,42,43,44]. Data that have shown efficacy have arisen mostly from animal-based experimentation [12,13,14,15]. Of these studies, anti-epileptic efficacy ranged from 3 to 4 months for older studies to nine months for the most recent study published in 2023. As such, the durability of graft-mediated seizure suppression in humans is yet unclear.

### 1.6. HLA-Matched iPSC Banks

Another methodology under investigation is to create a bank of clinical iPSC lines that are homozygous for HLA class I and class II genes that can be matched to recipients and differentiated when required. While sourcing adequate cell samples to create iPSCs from a large number of random donors may be impractical, one study used computer modeling to determine the minimum number of HLA donors required to reach such combinations for the United Kingdom (UK) population before searching the bone donor registry for these highly optimal HLA types and finding 58% of the types required, providing a theoretical zero mismatch for 95% of the UK population [26]. This strategy has since been implemented with success in research settings in Japan [27], Spain [51], Saudi Arabia [52], and South Korea [53], with the creation of HLA-matched iPSC banks from relatively small numbers of distinct cell lines that cover 6–51% of the respective country-wide populations. 

While these haplobank strategies mitigate the risk of acute rejection associated with unmatched allografting, and may be more cost-effective than culturing cortical inhibitory interneurons from autologous iPSCs, inherent challenges to this approach include finding donors for patients with HLA combinations that are more difficult to match [28], especially in more diverse populations such as the United States, and manufacturing cellular therapeutics from a wide array of resulting HLA-typed iPSCs [54]. In addition, tumorigenicity in iPSCs can arise if genetic mutations arise during the culture of iPSCs if iPSCs retain reprogramming factors and if iPSCs are not fully differentiated to final products. While whole-exome/whole-genome sequencing performed after reprogramming was originally time- and cost-prohibitive, recent developments have increased its viability in detecting genetic mutations [27,52]. However, the clinical significance of many such gene mutations has not yet been determined [27,55]; debate exists over the use of lines with mutations within disease-related Online Mendelian Inheritance in Man genes, and the significance of mutations in cancer-related genes that did not produce a carcinogenic phenotype in donors remaining unknown.

Differences between the implementations in different countries included reprogramming methods to generate iPSC lines, ranging from traditional viral vector methodologies utilizing preserved umbilical cord cells [51] to newer non-viral methodologies utilizing peripheral blood mononuclear cells or umbilical cord cells [27,52,55]. Such newer episomal plasmid-based reprogramming methods have provided a reliable yield of 0.01–0.06% [27,56], with benefits over retroviral methodologies of cost-effectiveness, ease of plasmid sequence removal after reprogramming, reduction in labor in handling infectious viruses with the inherent difficulty of modifying their programming factors, and reduction in the risk of transgenic sequence insertion [52].

### 1.7. Induction Techniques

Generation of human neurons from existing human stem cells has also advanced in capacity, reliability, and scope since 2007, when it was found that the application of a specific inhibitor decreased dissociation-induced apoptosis in human embryonic stem cells [54]. In the following years, several groups succeeded in generating >90% homogenous cortical interneuron precursors from human stem cells. A recent notable advancement has been the usage of a spinner culture over a static culture during induction, yielding increased precursors through the more efficient removal of wastes and delivery of nutrients [48]. Deriving specific interneuron subtypes most useful for cell therapy is under development, with the first direct induction of stem cells using direct genetic modification or expression of genetic factors, and more recent induction techniques utilizing chemical factors. Yields of GABAergic identities have ranged from 30 to 85%, and challenges have emerged in the risk of possible unintentional insertions or modifications to the genome, derivation of specific fast-spiking subtypes, maturation times, and expression of genes required for full natural function [57,58]. Chemical-based approaches to induction and maturation can generate interneurons with authentic phenotypes on a large scale, and methods that generate a synchronized population of early postmitotic interneurons have developed that show safety without continued proliferation and with maximal integration into the host with their avid migratory properties [47]. While more mature interneurons than those generated by chemical methods may be required for some applications, the developmental stage reached is suitable for most applications, and the generation of interneurons on a large scale without insertional mutagenesis is a potential upside.

While the specifics of induction techniques are beyond the scope of this review, in brief, the approach to generate inhibitory interneurons from human pluripotent stem cells can be divided in to the signaling molecule induction strategy and direct genetic reprogramming strategy [59]. Signaling molecule induction involves providing signaling molecules to recapitulate the neurodevelopmental process, requiring a longer time but recapitulating the native process, allowing for the study of pathogenetic mechanisms. Direct genetic manipulation employs exogenous transcription factors to bypass normal developmental pathways for significantly faster therapeutic identification with the additional benefit of the ability to differentiate somatic cells to inhibitory interneurons at the cost of possible differences to in vivo inhibitory interneuron counterparts and generation of confounders during analysis. Current challenges include the development of culture processes capable of obtaining more mature inhibitory interneurons, especially fast-spiking parvalbumin-expressing neurons.

#### Hypoimmunogenic iPSCs

Using hypoimmunogenic inhibitory interneurons genetically engineered to knock out one or more HLA class I/II genes represents a promising balance of the clinical need to prevent graft rejection with the scalability and rapid deployment offered by ‘off-the-shelf’ cell therapies. Early attempts at genetic engineering of hypo-immune iPSCs knocked out beta-2 microglobulin (B2M), the structural domain shared by all class I molecules [60]; however, the expression of either HLA-C [61,62] or other non-classical major histocompatibility complex (MHC) class I proteins such as HLA-E or HLA-G fused with B2M [63] is necessary to prevent natural killer (NK) cell-mediated transplant rejection. One study used the CRISPR-Cas9 system to knock out HLA-A and HLA-B expression with the concomitant knockout of HLA class II expression via targeting the class II major histocompatibility complex transactivator, as well as the upregulation of immune tolerance-promoting ligands such as programmed death-ligand 1, HLA-G, and CD47 to trigger anergy in T cells, NK cells, and macrophages, respectively, in a human pluripotent stem cell line with the preservation of differentiation capabilities [64].

At present, there is only one ongoing clinical trial of hypoimmunogenic iPSC-derived cell therapies, an open-label pilot study of pancreatic endodermal cell transplantation in patients with type 1 diabetes [65]. Notably, graft survival up to 40 weeks’ post-transplantation was recently reported in a rhesus macaque model of type 1 diabetes using cell grafts with MHC class I/II knockout combined with CD47 overexpression [66]. Assessment of the safety of this approach will be critical, as transplanting hypoimmunogenic cells into the CNS harbor risks generating immune-evasive teratomas or malignant tumors as well as harboring viruses without generating immune responses [16]. 

Limitations of hypoimmunogenic iPSCs include potential for off-target effects with insertional mutagenesis [60]. While this potential has been seen to be manifested in the experimental creation of hypoimmunogenic hPSCs [64], it did not impact the growth rate or differentiation of the pluripotent cell lines, and other studies have shown that modifying the genome to create hypoimmunogenic stem cells has less safety concerns than reprogramming it to create iPSCs [60,61]. A strategy recently developed to prevent tumorigenesis is the disruption of the uridine monophosphate synthetase-encoding gene to make growth dependent on an external and controllable supply of uridine [16].

### 1.8. MHC-Matching Controversy and Procedural Trauma

The host response to implanted foreign entities is one of the major challenges facing iPSC usage in epilepsy and other CNS disorders where iPSCs have been trialed, such as PD and HD [67]. While autologous iPSC cultures are the gold standard for mitigated immunogenicity, the associated challenges of cost and labor have prevented widespread trials and adoption. In other approaches, MHC-matching and hypoimmunogenic iPSCs have shown promise, with MHC-matching iPSC banks rising to the forefront given their promise of reduced immunogenicity while also minimizing the potential risk of neoplasia posed by hypoimmunogenic iPSC products [16]. However, the reduced immunogenicity of MHC-matched iPSC donors found in some studies [68] has been challenged by others [69], with a study in non-human primates showing a delayed sub-acute rejection, even in MHC-matched donors and recipients, requiring further optimization for safe grafting without risk of graft rejection.

The surgical approach and protocol for CNS implantation of stem cell therapies are of crucial importance to the survival of the graft. Inflammation induced by the surgical procedure, termed “needle trauma” for its similarity in pathophysiology to traumatic brain injury, is one of the factors in the likelihood of graft rejection [70] and may play a confounding role in the rejection of MHC-matched grafts [69]. This phenomenon is also seen clinically after insertional trauma from intraparenchymal leads to neurostimulation. Disruption of the blood–brain barrier (BBB), cells, capillaries, and the extracellular matrix allows for the “extravasation of inflammatory plasma proteins, a decrease in focal oxygen/nutrient delivery, microglial activation, mitochondrial dysfunction, increased oxidative stress and the accumulation of neurotoxic products in the brain parenchyma” [71]. Inhibiting the innate immune response to needle trauma involves several different approaches, one of which consists of applying the tools gleaned from research into inhibiting traumatic brain-injury-induced inflammation to the cell graft procedure, with its similar mechanism of innate inflammation [72]. In addition, co-implantation of autologous T-reg cells has been shown to substantially reduce surgical procedure-induced inflammation and improve therapeutic outcomes of iPSC implants in PD in rat models [70]. Finally, delaying cell insertion for over an hour after cannula insertion may allow for trauma-induced acutely released inflammatory cytokines to dissipate and increase cell graft survival [72].

## 2. Discussion

Drug-resistant epilepsy poses a challenge for treatment and therapy. The current standard of care [73] is to evaluate patients with severe or disabling seizures for ablative therapy before considering several last-line therapies (Table 1) for those that do not qualify. Inhibitory interneuron therapy may prove to be a useful addition to such last-line therapies, with different benefits and challenges that will become more elucidated as the therapy is developed. 

Our review complements previous reviews on the topic of iPSC use for neurologic disease by a novel focus on advances in iPSC use for drug-resistant epilepsy and overcoming the graft rejection of implanted cells. While other reviews have focused primarily on overcoming the immune response for cell therapy in PD and HD [67], or were older reviews surveying the overall usage of inhibitory interneurons for several neurologic diseases [8], our review updates the literature with the most recent findings on developments in inhibitory interneuron usage for drug-resistant epilepsy and the approaches to overcoming graft rejection. Our review complements a 2018 review by Zhu et al. [15] with the most recent findings and lessons from newer animal experimentations, including chemical induction, human Neurona clinical trials, developments in national HLA haplotype bank creation, and innovations in reducing “needle trauma” rejection, among others.

## 3. Conclusions

Usage of induced pluripotent stem cells for disease therapy has shown promise in challenging pathologies of different organ systems [35]. Drug-resistant epilepsy is one such disease with few effective, sustainable treatments [4]. While stem cell therapy has had emerging success in the treatment of other central nervous system pathologies, including Parkinson’s disease, Huntington’s disease, and stroke, one of its greatest challenges has remained the immunologic reaction against surgical procedures and implanted materials [68]. Extension of stem cell therapies to drug-resistant epilepsy will require surmounting the challenge of immunogenicity. A multitude of different approaches are under development, and more time will be required to differentiate an optimal path forward for the field.

## Figures and Tables

**Table 1 brainsci-14-01027-t001:** Summary of benefits and challenges of last-line therapies for DRE.

Intervention	Benefits	Challenges
Responsive cortical stimulation [RNS] [74,75,76]	Increasing benefits over time; positive cognitive side effects.	FDA-approved for ages 18 and older with focal epilepsy with 1–2 foci; more implantation-related complications; continued anti-seizure medications required.
Vagus nerve stimulation [VNS] [74,75,76]	FDA-approved for ages 4 and older with focal epilepsy; possibly effective for generalized epilepsy; increasing benefits over time.	Less effective than RNS or DBS at one year post-implantation; stimulation-related side effects; continued anti-seizure medications required.
Deep brain stimulation [DBS] [74,75,76]	Increasing benefits over time.	FDA-approved for ages 18 and older with focal epilepsy; negative cognitive side effects; more implantation-related complications; continued anti-seizure medications required.
Ketogenic dietary therapy [77]	Can be implemented earlier on in epilepsy management; modern implementations are more practical for adherence; likely similar efficacy in children and adults; is especially effective in several conditions.	Contraindicated in several specific fatty acid metabolic disorders; initial gastrointestinal distress; hyperlipidemia; renal calculi; height deceleration in children; continued anti-seizure medications required.
Inhibitory interneuron therapy	No reliance on devices for seizure control; disease-modifying treatment with possible curative applications.	Experimental: no FDA-approved therapies; immune rejection remains major challenge currently requiring immune suppression; potential for mutagenesis; currently unknown duration of effect.

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
