# Peer review of "Overcoming Graft Rejection in Induced Pluripotent Stem Cell-Derived Inhibitory Interneurons for Drug-Resistant Epilepsy"

_brainsci, 2024, doi:10.3390/brainsci14101027_

Round 1

Reviewer 1 Report

Comments and Suggestions for Authors

The manuscript by Beaudreault C and colleagues offers a comprehensive and insightful review on using the iPSC-derived interneurons to assist the treatment in DRE. It mainly focuses on then immune response related issues and proposed a thorough coverage of current approaches and potential solutions. The content is well-written covering key points with clinical evidence, bringing insights in understanding the clinical application and potential pitfalls. The manuscript is suitable for publication with some minor changes:

1: The learnings from cell-therapy in PD and HD were used as evidence for evaluating the potential can applicability into DRE treatment, however, epilepsy involves disrupted excitatory and inhibitory balance and may have different neural circuit from PD and HD. Limitations should be addressed or more evidence needs to be provided to further justify the benefits of iPSC-based cell therapy for DRE treatment.

2: In the section of hypoimmunogenic iPSCs, there is a lack of details regarding whether this approach could potentially introduce off-target effect with unintended gene alterations. Limitations of this approach should be discussed further. Also, whether the knock-out HLA genes would have detrimental consequence including tumorigenesis, or even affect the generation of interneurons should be discussed.

3: In the section of induction techniques, chemical-based approaches were discussed to generate interneurons with authentic phenotypes. However, more information is needed to support the proposed approach. For example, the comparison of timeline and cost, overall purity of final product, what chemical compounds were used, and comparison of induction efficiency.

4: More on the induction techniques, there is also lack of information to support the idea using chemical-based approaches to generate highly homogenous population of interneurons. Chemical-induction usually introduce more diversified cell populations. Generating >90% homogenous cortical interneuron precursors from human stem cells does not grant similar level of homogeneity in matured interneurons. Limitations and reliability of this approached should be discussed.

Author Response

1: The learnings from cell-therapy in PD and HD were used as evidence for evaluating the potential can applicability into DRE treatment, however, epilepsy involves disrupted excitatory and inhibitory balance and may have different neural circuit from PD and HD. Limitations should be addressed or more evidence needs to be provided to further justify the benefits of iPSC-based cell therapy for DRE treatment.

We appreciate the reviewer’s kind reminder to address limitations of this topic. On p. 6, the following text was added:

“Limitations of current allogeneic transplantation in epilepsy include the experimental nature of the procedure, with fewer evidence or ongoing trials [49] than in PD or HD [20-24,40-44]. Data that has shown efficacy has arisen mostly from animal-based experimentation [12-15]. Of these studies, anti-epileptic efficacy ranged from 3-4 months for older studies to nine months for the most recent study published in 2023. As such, the durability of graft-mediated seizure suppression in humans is yet unclear.”

2: In the section of hypoimmunogenic iPSCs, there is a lack of details regarding whether this approach could potentially introduce off-target effect with unintended gene alterations. Limitations of this approach should be discussed further. Also, whether the knock-out HLA genes would have detrimental consequence including tumorigenesis, or even affect the generation of interneurons should be discussed.

We also appreciate the reviewer’s request for limitations discussion with regards to off-target effects of inducing hypoimmunogenic iPSCs. On p. 8, the following text was added:
“Limitations of hypoimmunogenic iPSCs include potential for off-target effects with insertional mutagenesis [59]. While this potential has been seen manifested in experimental creation of hypoimmunogenic hPSCs [63], it did not impact the growth rate or differentiation of the pluripotent cell lines, and other studies have shown that modifying the genome to create hypoimmunogenic stem cells has less safety concerns than reprogramming it to create iPSCs [59, 60]. A strategy recently developed to prevent tumorigenesis is the disruption of the uridine monophosphate synthetase-encoding gene to make growth dependent on an external and controllable supply of uridine [16].”

3: In the section of induction techniques, chemical-based approaches were discussed to generate interneurons with authentic phenotypes. However, more information is needed to support the proposed approach. For example, the comparison of timeline and cost, overall purity of final product, what chemical compounds were used, and comparison of induction efficiency.

While we reviewed the differences between the two broad categories of induction techniques, we believe that more in-depth details about the process would be beyond the scope of our review on overcoming graft rejection. The following text was added to p. 7 to address reviewer points #3 and #4:
“While the specifics of induction techniques is beyond the scope of this review, in brief the approach to generate inhibitory interneurons from human pluripotent stem cells can be divided in to the signaling molecule induction strategy and direct genetic reprogramming strategy [77]. Signaling molecule induction involves providing signaling molecules to recapitulate the neurodevelopmental process, requiring a longer time but recapitulating the native process allowing for study of pathogenetic mechanisms. Direct genetic manipulation employs exogenous transcription factors to bypass normal developmental pathways for significantly faster therapeutic identification with the additional benefit of the ability to differentiate somatic cells to inhibitory interneurons, at the cost of possible differences to in vivo inhibitory interneuron counterparts and generation of confounders during analysis. Current challenges include development of culture processes capable of obtaining more mature inhibitory interneurons, especially fast-spiking parvalbumin-expressing neurons.”

4: More on the induction techniques, there is also lack of information to support the idea using chemical-based approaches to generate highly homogenous population of interneurons. Chemical-induction usually introduce more diversified cell populations. Generating >90% homogenous cortical interneuron precursors from human stem cells does not grant similar level of homogeneity in matured interneurons. Limitations and reliability of this approached should be discussed.

Please refer to revised manuscript p. 7: While we reviewed the differences between the two broad categories of induction techniques, we believe that more in-depth details about the process would be beyond the scope of our review on overcoming graft rejection.
Please also see response to reviewer point #3, above, for text which addresses this point.

Reviewer 2 Report

Comments and Suggestions for Authors

Very interesting review. It is clearly written and describes relevant pros and cons related to the use of iPSC-derived inhibitory interneurons for drug-resistant epilepsy. The quoted literature should provide a good source of relevant information for general readers as well as active scientists working in the field of interest.

Author Response

Comments and Suggestions for Authors

Very interesting review. It is clearly written and describes relevant pros and cons related to the use of iPSC-derived inhibitory interneurons for drug-resistant epilepsy. The quoted literature should provide a good source of relevant information for general readers as well as active scientists working in the field of interest.

Author response: Thank you.

Reviewer 3 Report

Comments and Suggestions for Authors

-It is better to use the full term 'Human Leukocyte Antigen (HLA)' the first time it appears in the abstract before using the abbreviation.

-In the introduction, authors may talk about the graft rejection.

-Authors may include animal model studies in the 'Allogeneic Transplantation in Epilepsy' section.

Author Response

Comments and Suggestions for Authors

-It is better to use the full term 'Human Leukocyte Antigen (HLA)' the first time it appears in the abstract before using the abbreviation.

Author response: see p. 1

“… Similarly, in HD patients a pilot study investigating transplantation of medial ganglionic eminence (MGE) derived cells demonstrated degrees of alloimmunization and graft rejection[23],24 and a phase II multicenter trial based on the previous pilot study reported many serious adverse events for which graft rejection was implicated[24]. Graft rejection, primarily driven by the absence of histocompatibility in human leukocyte antigen (HLA) between graft and host[25], remains a challenge for cell-based therapies for CNS disorders including DRE.”

-In the introduction, authors may talk about the graft rejection.

Author response: see p. 2: “With clinical trials underway, the challenge of maintaining long term graft viability and avoiding chronic rejection remains. The aim of the present literature review is to evaluate various strategies for modulating the host immune response to inhibitory interneuron transplantation.”

-Authors may include animal model studies in the 'Allogeneic Transplantation in Epilepsy' section.

Author response: see p. 5: “Limitations of current allogeneic transplantation in epilepsy include the experi-mental nature of the procedure, with fewer evidence or ongoing trials [49] than in PD or HD [20-24,40-44]. Data that has shown efficacy has arisen mostly from animal-based experimentation [12-15]. Of these studies, anti-epileptic efficacy ranged from 3-4 months for older studies to nine months for the most recent study published in 2023. As such the durability of graft-mediated seizure suppression in humans is yet unclear.”

Reviewer 4 Report

Comments and Suggestions for Authors

The problem of drug-resistant epilepsy is actual, despite the development of new generations of antiepileptic drugs. Surgical methods are also being modified and updated, but the problem under consideration is still far from being resolved. The topic of this narrative review is interesting from a scientific and practical point of view.

The manuscript needs a small technical revision (see the MDPI template): remove the list of abbreviations; links to cited publications should be placed in square brackets.

I recommend that the authors add a Discussion section in which to explain the advantages and limitations of the method of cell therapy for drug-resistant epilepsy using induced interneurons derived from induced pluripotent stem cells, compared with previously known methods of surgical treatment of this disease (please add a table or figure). Are there any previously published systematic reviews or meta-analyses on this topic? How does your narrative review differ from or complement previously published reviews?

Author Response

The manuscript needs a small technical revision (see the MDPI template): remove the list of abbreviations; links to cited publications should be placed in square brackets.

We are grateful for the reviewer’s kind suggestion for technical revisions; these have now been addressed. Please see p. 1 and in-text

I recommend that the authors add a Discussion section in which to explain the advantages and limitations of the method of cell therapy for drug-resistant epilepsy using induced interneurons derived from induced pluripotent stem cells, compared with previously known methods of surgical treatment of this disease (please add a table or figure). Are there any previously published systematic reviews or meta-analyses on this topic? How does your narrative review differ from or complement previously published reviews?

The reviewers are right to request comparison of cell-based surgical interventions with established surgical techniques for treating drug-resistant epilepsy. We have added the requested table; please see revised manuscript p. 9-10. A discussion addressing previous literature reviews on the topic is also added to the same section.

Reviewer 5 Report

Comments and Suggestions for Authors

This review article focuses on cell-based therapies for central nervous system disorders. The authors initially proposed in the title to review graft rejection and the use of immunosuppressive approaches for post-transplantation outcomes of this therapy specifically in epilepsy patients. The topic of drug-resistant epilepsy is an important area of research that remains in high demand. However, the paper has little focus on epilepsy and instead discusses the success of cell-based therapy in various brain disorders, such as Parkinson’s' and Huntington's diseases, discussing issues with graft rejections there. Therefore,  the authors should revise the title accordingly to reflect the broader scope of the review, stating cell-based therapies in brain pathologies; otherwise, they should conduct more work on the topic in epilepsy. One can understand that the idea of implanting iPSC-derived inhibitory interneurons might not be well-developed in the field, but there are other available directions to treat drug-resistant epilepsy that should be discussed and compared to keep the focus of this review on epilepsy. From the current review content, it is not even clear whether there have been clinical trials on grafting inhibitory interneurons in epilepsy patients.

Also, the paper lacks illustrations. The authors have to consider including schemes or tables to visualise the key ideas of their review paper.

Comments on the Quality of English Language

Minor language check is required. 

Author Response

Comments and Suggestions for Authors

This review article focuses on cell-based therapies for central nervous system disorders. The authors initially proposed in the title to review graft rejection and the use of immunosuppressive approaches for post-transplantation outcomes of this therapy specifically in epilepsy patients. The topic of drug-resistant epilepsy is an important area of research that remains in high demand. However, the paper has little focus on epilepsy and instead discusses the success of cell-based therapy in various brain disorders, such as Parkinson’s' and Huntington's diseases, discussing issues with graft rejections there. Therefore,  the authors should revise the title accordingly to reflect the broader scope of the review, stating cell-based therapies in brain pathologies; otherwise, they should conduct more work on the topic in epilepsy.

One can understand that the idea of implanting iPSC-derived inhibitory interneurons might not be well-developed in the field, but there are other available directions to treat drug-resistant epilepsy that should be discussed and compared to keep the focus of this review on epilepsy. From the current review content, it is not even clear whether there have been clinical trials on grafting inhibitory interneurons in epilepsy patients.

Author response: We respectfully decline to revise the title of the review, given that the present work deals with graft rejection in the context of drug-resistant epilepsy; while we discuss research in PD and HD, that is because these fields have more available clinical trial data; however, the scope is focused because of the unique properties of inhibitory interneurons as a cell-based therapy for drug-resistant epilepsy, which we also discuss at length. It would be inappropriate to revise the title given that this is a mini-review which concerns itself with a chosen pathology.

Regarding the discussion of in-human clinical trials in epilepsy, please see p. 5: “Utilization of allogeneic transplantation of off-the-shelf cell lines is now in Phase I/II clinical trials[48], and represents a promising approach to cell-based therapy for epilepsy patients. Due to the risk of graft rejection, the trial protocol uses perioperative and postoperative immune-suppressing therapy, with the option to wean and discontinue therapy after one year follow-up[49]. Opportunistic infections, endocrinologic dysregulation and metabolic syndrome, as well as increased risk of developing certain malignancies, are all risks undertaken by any patient on immunosuppressive therapy, and the main adverse events reported by Neurona investigators which are attributable to the trial itself have been due to patients’ immune-suppressing regimens[50]. Notably, although the two NRTX-1001 trial patients with the longest follow-up reportedly maintained graft-associated reductions in seizure frequency >95% at >1 year post-transplant,, imaging-based follow-up is planned for up to two years post-transplant and annual office visits supplemented by quarterly phone calls are planned for up to thirteen years post transplant. These long-term follow up studies will provide important evidence for or against the presence of chronic graft rejection in allogeneic cell lines of inhibitory interneurons.

Limitations of current allogeneic transplantation in epilepsy include the experimental nature of the procedure, with fewer evidence or ongoing trials [49] than in PD or HD [20-24,40-44]. Data that has shown efficacy has arisen mostly from animal-based experimentation [12-15]. Of these studies, anti-epileptic efficacy ranged from 3-4 months for older studies to nine months for the most recent study published in 2023. As such the durability of graft-mediated seizure suppression in humans is yet unclear.”

Also, the paper lacks illustrations. The authors have to consider including schemes or tables to visualise the key ideas of their review paper.

Author response: See p. 8, Table 1 (following page):

Table 1. Summary of Benefits and Challenges of Last-Line Therapies for DRE.

Benefits

Challenges

Responsive cortical stimulation [RNS] [73-75]

Increasing benefits over time; positive cognitive side effects;

FDA approved for ages 18 and older with focal epilepsy with 1-2 foci; more implantation-related complications; continued anti-seizure medications required

Vagus Nerve Stimulation [VNS] [73-75]

FDA approved for ages 4 and older with focal epilepsy; possibly effective for generalized epilspy; Increasing benefits over time;

Less effective than RNS or DBS at one year post-implantation; stimulation-related side effects; continued anti-seizure medications required

Deep Brain Stimulation [DBS] [73-75]

Increasing benefits over time;

FDA approved for ages 18 and older with focal epilepsy; negative cognitive side effects; more implantation-related complications; continued anti-seizure medications required

Ketogenic Dietary Therapy[76]

Can be implemented earlier on in epilepsy management; modern implementations are more practical for adherence; likely similar efficacy in children and adults; is especially effective in several conditions.

Contraindicated in several specific fatty acid metabolic disorders; initial gastrointestinal distress; hyperlipidemia; renal calculi; height deceleration in children; continued anti-seizure medications required

Inhibitory Interneuron Therapy

No reliance on devices for seizure control; disease-modifying treatment with possible curative applications.

Experimental: no FDA approved therapies; immune rejection remains major challenge currently requiring immune suppression; potential for mutagenesis; currently unknown duration of effect.

Round 2

Reviewer 4 Report

Comments and Suggestions for Authors

I thank the authors for responding to my comments and modifying the manuscript.

Please add a link to Table 1 in the text.

In this table, add the name of the first column.

Remove the bold text in this table, except for the column names.

Author Response

Comment 1: I thank the authors for responding to my comments and modifying the manuscript.

Response 1: We thank the reviewer for their gracious feedback

Comment 2: Please add a link to Table 1 in the text.

Response 2: The following in-text citation was added to the discussion section on page 8 of 12:

Drug-resistant epilepsy poses a challenge for treatment and therapy. The current standard of care [72] is to evaluate patients with severe or disabling seizures for ablative therapy before considering several last-line therapies (Table 1)

Comment 3: In this table, add the name of the first column.

Response 3: The column name “Intervention” has been added to the first column

Comment 4: Remove the bold text in this table, except for the column names.

Response 4: As requested, bold text has been removed from Table 1, with the exception of column names. See revised Table 1, below (also found on page 8 of 12 of the manuscript):

Table 1. Summary of Benefits and Challenges of Last-Line Therapies for DRE.

Intervention

Benefits

Challenges

Responsive cortical stimulation [RNS] [73-75]

Increasing benefits over time; positive cognitive side effects;

FDA approved for ages 18 and older with focal epilepsy with 1-2 foci; more implantation-related complications; continued anti-seizure medications required

Vagus Nerve Stimulation [VNS] [73-75]

FDA approved for ages 4 and older with focal epilepsy; possibly effective for generalized epilepsy; Increasing benefits over time;

Less effective than RNS or DBS at one year post-implantation; stimulation-related side effects; continued anti-seizure medications required

Deep Brain Stimulation [DBS] [73-75]

Increasing benefits over time;

FDA approved for ages 18 and older with focal epilepsy; negative cognitive side effects; more implantation-related complications; continued anti-seizure medications required

Ketogenic Dietary Therapy[76]

Can be implemented earlier on in epilepsy management; modern implementations are more practical for adherence; likely similar efficacy in children and adults; is especially effective in several conditions.

Contraindicated in several specific fatty acid metabolic disorders; initial gastrointestinal distress; hyperlipidemia; renal calculi; height deceleration in children; continued anti-seizure medications required

Inhibitory Interneuron Therapy

No reliance on devices for seizure control; disease-modifying treatment with possible curative applications.

Experimental: no FDA approved therapies; immune rejection remains major challenge currently requiring immune suppression; potential for mutagenesis; currently unknown duration of effect.